# Preparation and Properties of Crosslinked Quaternized Chitosan-Based Hydrogel Films Ionically Bonded with Acetylsalicylic Acid for Biomedical Materials

**DOI:** 10.3390/md22100450

**Published:** 2024-09-30

**Authors:** Jingjing Zhang, Linqing Wang, Yingqi Mi, Fang Dong, Zhanyong Guo

**Affiliations:** 1Key Laboratory of Coastal Biology and Bioresource Utilization, Yantai Institute of Coastal Zone Research, Chinese Academy of Sciences, Yantai 264003, China; jingjingzhang@yic.ac.cn (J.Z.); lqwang@yic.ac.cn (L.W.); fdong@yic.ac.cn (F.D.); 2University of Chinese Academy of Sciences, Beijing 100049, China

**Keywords:** acetylsalicylic acid, antibacterial activity, anti-inflammatory activity, quaternized chitosan, hydrogel films

## Abstract

The aim of the current study is to develop chitosan-based biomaterials which can sustainably release acetylsalicylic acid while presenting significant biological activity. Herein, an innovative ionic bonding strategy between hydroxypropyl trimethyl ammonium chloride chitosan (HACC) and acetylsalicylic acid (AA) was proposed, skillfully utilizing the electrostatic attraction of the ionic bond to achieve the controlled release of drugs. Based on this point, six crosslinked *N*-[(2-hydroxy-3-trimethylammonium)propyl]chitosan acetylsalicylic acid salt (CHACAA) hydrogel films with varying acetylsalicylic acid contents were prepared by a crosslinking reaction. The results of ^1^H nuclear magnetic resonance spectroscopy (^1^H NMR) and scanning electron morphology (SEM) confirmed the crosslinked structure, while the obtained hydrogel films possessed favorable thermal stability, mechanical properties, and swelling ability. In addition, the drug release behavior of the hydrogel films was also investigated. As expected, the prepared hydrogel films demonstrated the capability for the sustainable release of acetylsalicylic acid due to ion pair attraction dynamics. Furthermore, the bioactivities of CHACAA-3 and CHACAA-4 hydrogel films with acetylsalicylic acid molar equivalents of 1.25 and 1.5 times those of HACC were particularly pronounced, which not only exhibited an excellent drug sustained-release ability and antibacterial effect, but also had a higher potential for binding and scavenging inflammatory factors, including NO and TNF-α. These findings suggest that CHACAA-3 and CHACAA-4 hydrogel films hold great potential for applications in wound dressing, tissue engineering scaffolds, and drug carriers.

## 1. Introduction

Hydrogels are polymeric materials with a three-dimensional crosslinked network structure that have a high affinity for water, yet remain insoluble because of the existence of physical or chemical bonds between polymer chains [1,2,3]. The porous structure enables hydrogels to absorb abundant amounts of aqueous solvents, thereby increasing their size and swelling [4,5]. The hydrophilic quality of hydrogels, combined with the physicochemical similarity to biological cell matrices, contributes to their excellent mechanical stability and biocompatibility [6]. Recently, hydrogels, especially carbohydrate-based hydrogels, have garnered significant interest among biomaterial scientists and have been found diverse applications in fields such as food science [7], biomedicine [8,9], wastewater treatment [10,11], and soft machinery [12].

Chitosan (CS) is a natural polysaccharide carbohydrate obtained by more than 50% deacetylation of chitin, which is primary extracted from crustacean shells and is one of the most abundant natural polymers [13,14]. Chitosan holds significant potential for various applications due to its non-toxicity, biocompatibility, and biodegradability [15]. In particular, chitosan, as a biopolymer with a unique chemical structure and abundant availability, is an ideal candidate for hydrogel materials [16,17]. However, it suffers from weak bioactivity and poor solubility in neutral and alkaline conditions, and its inherent strong hydrogen bonds also make chitosan difficult to be directly used as a biomaterial for in situ gelling [15,18]. The modification of these characteristics is a necessity for realizing the high-value utilization of chitosan. Accordingly, the chemical modification of chitosan involving amino and hydroxyl groups of glucosamine monomers by crosslinking and grafting reactions is a recommended approach to obtain an effective hydrogel material over a wide pH range [19,20]. Furthermore, the introduced functional binding groups on chitosan chain can synergistically strengthen the hydrogel network and enhance the bioactivity, thus enabling advancements in biomedical applications such as drug delivery, wound healing, biosensors, bone regeneration, tissue engineering, and bio-membrane fabrication [21,22,23].

Hydroxypropyl trimethyl ammonium chloride chitosan (HACC), a quaternized chitosan derivative polymer, exhibits excellent aqueous solubility across a wide pH range [24]. Importantly, HACC carries strong positive charges due to the abundant amine groups along its backbone, which can combine with active drugs through ionic bond interaction, thereby achieving a controlled release function and improving the concentration and residence time of drugs at the focal site. For example, Jin et al. had used an electrostatic interaction between HACC modified with D-mannose (M-N-2-HACC) and carboxymethyl chitosan containing *N*-acetyl-L-cysteine (N-CMCS) to construct a mucoadhesive carrier for drug delivery. It had been reported to possess high hydrophilicity, enduring drug release, stability, and mucosal adhesion, which could be used as a promising candidate for mucosal delivery [25]. Additionally, previous studies have demonstrated that HACC possessed low toxicity, high safety, and exhibited significant antimicrobial activity against various bacteria and fungi, making it an excellent candidate for the fabrication of multifunctional hydrogels [26]. Therefore, the development of hydrogel materials based on HACC may offer a unique drug delivery platform with multifunctional qualities and the novel design strategy based on electrostatic interactions among transporters and active drugs may provide a promising approach for efficient drug delivery.

Acetylsalicylic acid (AA), commonly known as aspirin, is a widely used non-steroidal anti-inflammatory drug in clinical medicine [27,28]. Its anti-inflammatory mechanism primarily involves the inhibition of TNF-α, cyclooxygenase enzymes (COX-1, COX-2), and the inducible nitric oxide synthase [29,30]. In addition, acetylsalicylic acid plays a key role in the process of NF-κB gene transcription, which can inhibit the production of pro-inflammatory cytokines such as IL-6, IL-8, and IL-1β [27,31]. These anti-inflammatory effects make acetylsalicylic acid effective at treating conditions such as fever, muscle and joint aches, headaches, and inflammation-related cancers [32]. However, prolonged use of acetylsalicylic acid can cause damage to the gastrointestinal tract, liver, kidneys, central nervous system, and increase the risk of irregular heartbeat [33,34]. Incorporating AA into biocompatible HACC through ionic bond interaction to achieve a controlled release function could help mitigate these risks. Specifically, this can not only give full play to its pharmacological effect, but also help to balance the blood drug concentration, reduce the frequency of administration, and weaken the stimulation of gastrointestinal mucosa [35,36]. Therefore, the exploration of HACC as a hydrogel sustained-release material is of great significance to developing new acetylsalicylic acid delivery methods.

In this contribution, crosslinked HACC hydrogel (CHACC) was fabricated between the hydroxyl group in HACC and epichlorohydrin, which can result in high mechanical strength. Acetylsalicylic acid was then incorporated into the hydrogel networks at various doses to obtain CHACAA hydrogel films. The electrostatic interaction between acetylsalicylic acid and CHACC contributed to a more controlled release ability. The chemical structure of CHACAA hydrogel films was confirmed by ^1^H NMR spectroscopy and SEM. Additionally, various physicochemical properties, including thermal analysis, water content, swelling degree, water solubility, biodegradation ratio, and mechanical behavior, were systematically evaluated. Furthermore, we investigated the release profiles and mechanisms of acetylsalicylic acid from different CHACAA hydrogel films in vitro at a pH of 7.4. Finally, the antibacterial ability and the anti-inflammatory property were mainly assessed. The results demonstrated that the incorporation of acetylsalicylic acid into the hydrogel film network allowed for sustained and controlled release, leading to long-term therapeutic effects. Moreover, the hydrogel films with acetylsalicylic acid exhibited enhanced antibacterial and anti-inflammatory performance. Consequently, CHACAA, as a multifunctional hydrogel, has the potential to be a promising candidate for treating infections in complex tissues and may have significant implications in biomedical material applications.

## 2. Results and Discussion

### 2.1. Characterization of Films

The hydrogel films were developed using the environmentally friendly biomass quaternized chitosan (HACC) as a matrix, epichlorohydrin as a crosslinker, and acetylsalicylic acid as a reinforcing filler. The detailed synthesis process was illustrated in Figure 1: Firstly, HACC was synthesized according to previous methods [37]. Then, HACC was crosslinked with epichlorohydrin to form a CHACC three-dimensional (3D) network under alkaline condition. During this reaction, glycerol was added to form hydrogen bonds with the hydrogel system. Finally, different doses of acetylsalicylic acid were added and a series of hydrogel films was attained, when the molar ratios between acetylsalicylic acid and quaternized chitosan varied from 0.75/1 up to 2/1. In the hydrogel 3D network structure, one part of acetylsalicylic acid anions was bound to CHACC by ionic bonding, while the other part was physically blended. The structures of all hydrogel films were confirmed by ^1^H nuclear magnetic resonance spectroscopy (Figure 1), scanning electron morphology (Figure 2), and thermal analysis (Figure 3). Their mechanical properties, water content, swelling degree, water solubility, and in vitro degradation were also analyzed, and the results were shown in Table 1 and Table 2.

#### 2.1.1. ^1^H Nuclear Magnetic Resonance Spectroscopy (^1^H NMR)

The chemical structures of CS, HACC, HACAA, and CHACAA were confirmed using ^1^H NMR spectral analysis, and the peak attributions were labeled in Figure 1. From the spectra, it could be seen that the pure CS showed signals at 3.6–3.9 ppm and 3.0 ppm, which were assigned to H3-H6 and H2 of the methine protons of the constitutional unit of chitosan [38]. Following the spectrum of HACC, in addition to the characteristic signals of chitosan, the absorbed signals of the quaternary ammonium salt group could be clearly observed. In the spectrum of HACAA, the insertion of acetylsalicylic acid was clearly evidenced with the newly appeared peaks at 6.8–7.7 ppm and 3.1 ppm, which were attributed to the hydrogen protons of the benzene ring and the methyl group, respectively [35]. Compared to HACAA, the ^1^H NMR spectra of CHACAA 1–6 presented a new peak at 3.4 ppm, which was attributed to the methylene hydrogen protons and methylidyne hydrogen protons of -CH_2_CH(OH)CH_2_-, proving that epichlorohydrin successfully reacted with HACC [39]. Also, the characteristic peaks of quaternary ammonium salt and acetylsalicylic acid groups were still observed, indicating that CHACAA was successfully synthesized.

#### 2.1.2. Scanning Electron Morphology (SEM) Analysis

The pore structures of CHACAA hydrogel films formed after the freeze-drying process were assessed visually using SEM analysis (Figure 2). Before water absorption, the CHACAA-1 hydrogel film was uniform and compact. In contrast, the hydrogel films exhibited a 3D microporous structure after absorbing water, with the macropores being flat in shape and ranging in size from 100 to 400 μm. This is because the water molecules infiltrate into the hydrogel films and enter the space between crosslinked HACC molecular chains, allowing the molecular chains to stretch out as far as possible. However, due to the connection of the crosslinking points, the HACC molecular chain will not break. The hydrogel films will swell and present obvious pores in the interior, and eventually show a highly interconnected porous construct. These interconnected pores create a favorable environment for the effective loading of drugs into the hydrogel films, and significantly affect the flow and release of the drugs.

#### 2.1.3. Thermal Analysis

The thermal properties of CHACAA hydrogel films were investigated by TGA and DTG curves, and the results are shown in Figure 3. The samples showed three-step degradation processes in range from 40 °C to 500 °C [19]. The first stage was observed from 60 °C to 140 °C, attributed to the loss of absorbed water and connected water in hydrogel films. At this stage, the changes in CHACAA 1-2 were slightly more obvious than those in CHACAA 3–6. The second phase of weight loss was in the range of 160 °C to 260 °C, which was due to the destruction of the hydrogel network structure by breaking the crosslinks (-CH_2_CH(OH)CH_2_-). It could also be found that different CHACAA hydrogel films exhibited different thermogravimetric behavior at this stage, that is, the maximum weight loss rates of the hydrogel films gradually decreased with the increase in acetylsalicylic acid content. Another weight loss was seen at around 270 °C, which was attributed to the decomposition of the remaining molecules of HACAA. The above observations were consistent with the findings reported by Li et al. [39]. Significantly, CHACAA-2, CHACAA-3, and CHACAA-4 reached the highest temperature (267 °C) at the maximum weight loss rate, indicating that these hydrogel films possessed the best thermal stability after the crosslinking reaction.

#### 2.1.4. Thickness, Density, and Mechanical Properties

Table 1 summarizes the thickness, density, and mechanical properties of CHACAA hydrogel films. Obviously, the thickness and density were positively correlated with the amount of acetylsalicylic acid. For example, CHACAA-6 with the highest content of acetylsalicylic acid exhibited the highest thickness and density, measuring 426.55 ± 4.69 μm and 1.11 ± 0.03 g/cm^3^, respectively. Mechanical properties are key factors affecting the application of hydrogels in the biomedical field since biomedical materials require resistance to stress and stretch-out caused by body movement [40]. The test results showed that all hydrogel films had favorable tensile strength (TS) and elongation at break (EB), owing to the 3D crosslinked network. In particular, CHACAA-3 showed the best mechanical properties, with TS and EB values of 11.83 ± 0.31 MPa and 105.27 ± 1.54%, respectively. This might because the blending of quaternized chitosan and acetylsalicylic acid in this ratio caused polymer chains to have more chain entanglement, and the higher applied tensile force was used to overcome the high amount of polymer chain entanglement, resulting in higher tensile strength and elongation at break. Studies have shown that the TS and EB of human skin tissue are in the range of 1–32 MPa and 17–207%. The CHACAA hydrogel films prepared in this paper were all in the range of mechanical properties suitable for human skin tissue, indicating a good mechanical basis for biomedical materials.

#### 2.1.5. Water Content, Swelling Degree, Water Solubility, and Biodegradation Ratio

The water content, swelling degree, water solubility, and biodegradation ratio were determined and the results are shown in Table 2. The water content of all hydrogel films was relatively stable, maintaining at about 7–8.5%. Among them, CHACAA-2 and CHACAA-3 had the higher water content, with values of 8.38 ± 0.28% and 8.26 ± 0.16%, respectively. In addition, CHACAA-3 showed the highest degree of swelling, reflecting a better ability to absorb and retain liquids in the network. The water solubility and degradation ratio of CHACAA hydrogel films determine its service cycle and directly affect the use function as medical materials. It can be seen from Table 2 that CHACAA-4, CHACAA-5, and CHACAA-6 possessed better water solubility, and the in vitro degradation ratio of all hydrogel films had a positive relationship with the concentration of acetylsalicylic acid, that is, with the increase in acetylsalicylic acid content, the degradation ratio of the hydrogel films increased. Probably these events were in line with the enhancement of the water solubility of these hydrogel films after embedding acetylsalicylic acid in the chitosan-based network. Therefore, appropriate cation–anion interaction between CHACC and AA could provide a stable network with an acceptable destruction rate.

### 2.2. In Vitro Release Studies

Since the CHACAA hydrogel films are intended to be prepared as medical material for tissue engineering, bio-adhesives, and wound dressing patch applications, the in vitro drug release experiment was performed at pH 7.4 to simulate the normal environment of bodily fluids. The graph of the cumulative percentage of drug release after some hours is depicted in Figure 4. The in vitro release of uncoated acetylsalicylic acid exhibited a “burst effect”, which might become the source of shortened action time, multiple and repeated dosing, and increased cytotoxicity of the liver and kidneys. CHACAA hydrogel films, by contrast, showed an obvious sustained release effect, and all hydrogel films followed the same pattern based on two different phases: the first phase consisted of a burst release up to 2 h, while the second phase corresponded to a sustained release process up to 24 h or even more. This phenomenon is in accordance with the literature results [41]. For example, Xu et al. investigated the release behavior of acetylsalicylic acid from the chitosan/β-sodium glycerophosphate/gelatin hydrogels. They found that the prepared hydrogels underwent an initial burst release phase followed by a sustained slow release stage [42]. According to the analysis, the burst release phenomenon was mainly caused by two reasons. On the one hand, AA molecules adsorbed on the surface of hydrogel films or that interacted weakly with CHACC could release in a short time, driven by the diffusion effect. On the other hand, hydrogel films could form hydrogen bonds when exposed to buffer solution, facilitating the free diffusion of the covered drug set in the most superficial regions [43]. In the sustained release stage, the drug was embedded in the hydrogel because of electrostatic attraction, so it took a long time for AA to be released through the gel layer in the medium. Of course, the test hydrogel films released acetylsalicylic acid in a different manner at different rates. CHACAA-1 had a sustained drug release effect, but the cumulative release of AA was only 40.11 ± 1.60% at 48 h. This behavior was mainly attributed to the strong interaction between the positively charged CHACC and the negatively charged AA. As for CHACAA-5 and CHACAA-6, the burst release was obvious due to the high content of acetylsalicylic acid and they were not an ideal choice for drug delivery materials. An obvious cumulative release related to the release of more than 50% of the drug in the first 12 h was observed for CHACAA-2, CHACAA-3, and CHACAA-4. Thenceforth, the AA release profiles converted to a slow-release step, in such a way that about 60% of the loaded drug was freely released within 48 h. Hence, they had advantages of controlling drug release and prolongating drug action time, and presented broad application prospects in the fields of controlled drug release materials.

### 2.3. Antibacterial Activity

Gram-positive bacteria (*Escherichia coli* ATCC 25922) and gram-negative bacteria (*Staphylococcus aureus* CMCC(B) 26003) were chosen as model bacteria to evaluate the antibacterial activity of CHACAA hydrogel films, while a blank group without a hydrogel sample was the control group. It could be intuitively seen from Figure 5 that a large number of bacterial colonies were formed on the agar plates of each strain in the control group. After being treated with CHACAA hydrogels, almost no bacteria survived on the culture dishes, indicating that CHACAA hydrogels maintained high antibacterial activity, and the antibacterial rate could reach 100%. This remarkable inhibition behavior is mainly due to the fact that positively charged quaternary ammonium groups can interact with the negatively charged bacterial biofilm through electrostatic adhesion and efficiently damage the bacterial biofilms [44]. (Note: since no new colonies were formed on the petri dishes treated with all hydrogel films, only the photographs of bacterial colonies of the blank group and one group of samples are shown in Figure 5 for convenient observation.)

### 2.4. Anti-Inflammatory Activity

Persistent chronic inflammation delays the body’s healing process and induces pain. In particular, the presence of bacterial infections leads to progressive destruction of tissue due to the accumulation of inflammatory cells and the secretion of excessive inflammatory cytokines [45]. Therefore, an adequate reducing inflammatory response is highly necessary. Here, the anti-inflammatory activity of CHACAA hydrogel films was established by measuring the effect on the release of NO and TNF-α from RAW 264.7 cells. The cytotoxicity test, of course, was first performed with this cell line in order to avoid false inflammatory activity produced by the death of the cells. The cytotoxicity results were depicted in Figure 6a. In the presence of CHACAA-1 and CHACAA-2, the viability of RAW 264.7 cells remained at ~100%, a fact which clearly indicates the lack of cytotoxicity. With the increase in acetylsalicylic acid content, the treated groups had a decrease in the percentage of viable cells. Among them, the survival rates of RAW 264.7 cells treated with CHACAA 4–6 dropped to ~80%. Nevertheless, the cell viability of all hydrogel samples remained at satisfactory values, higher than 80%, which indicated that the in vitro anti-inflammatory activity could continue to be evaluated.

The total amount of NO (expressed as a percentage) released by LPS-induced RAW 264.7 cells after treatment with CHACAA hydrogel film extracts is shown in Figure 6b. It could be concluded that all hydrogel films significantly reduced the release of NO compared with the positive control group. In particular, with the increase in acetylsalicylic acid, the released amount of NO decreased, which fully indicated that the presence of acetylsalicylic acid had a positive effect on improving the anti-inflammatory activity of the chitosan-based hydrogel material. Among them, the inhibition rates of CHACAA-1 and CHACAA-2 on the release of NO were about 45%, while CHACAA-3 and CHACAA-4 could inhibit more than 70% of NO under the same conditions.

The percentage expression of inflammatory cytokine TNF-α is shown in Figure 6c. The expression of pro-inflammatory cytokine TNF-α was significantly decreased in the CHACAA group as compared to the positive group, indicating that the prepared hydrogel films had a significant containment effect on the LPS-induced inflammation model in vitro. Moreover, CHACAA-3, CHACAA-4, CHACAA-5, and CHACAA-6 reduced TNF-α levels more effectively, with expression values of 13.44%, 15.00%, 12.12%, and 14.97%, respectively, and they were about three times as effective as CHACAA-1. These results confirmed the anti-inflammatory activity of chitosan-based hydrogels and demonstrated the advantages of incorporating acetylsalicylic acid into the hydrogel to effectively attenuate the hypersecretion of proinflammatory cytokines and increase drug bioavailability.

## 3. Materials and Methods

### 3.1. Materials

Chitosan with 60 kDa molecular weight and 76% deacetylation degree was purchased from Golden-Shell Pharmaceutical Co., Ltd., (Yuhuan, China). Acetylsalicylic acid was supplied from Sigma-Aldrich Chemical Corp. (Shanghai, China). Ethanol, sodium hydroxide, isopropanol, epichlorohydrin, glycerin, and hydrochloric acid were purchased from the Sinopharm Chemical Reagent Co., Ltd. (Shanghai, China). Glycidyl trimethyl ammonium chloride was acquired from Anhui Zesheng Technology Co., Ltd. (Anqing, China). All chemical solvents and reagents were analytical grade and used as received.

### 3.2. Preparation of Crosslinked Quaternized Chitosan-Based Hydrogel Films Containing Acetylsalicylic Acid

Firstly, hydroxypropyl trimethylammonium chloride chitosan quaternary ammonium salt (HACC) was obtained and the synthesis process was followed by the previous methods [37]. Then, 2.0 g of HACC was sufficiently dissolved in 100 mL of deionized water in a 250 mL round-bottomed flask with mechanical stirring, and the pH value of the solution was adjusted to 10 by adding 5% sodium hydroxide solution. Subsequently, 4 mL of epichlorohydrin was added for the cross-linking reaction. The pH of the reaction mixture was maintained at about 10 during this period. After the reaction for 4 h at 70 °C, the pH of the reaction system was adjusted to 7 by adding 10% hydrochloric acid solution. After that, the reaction solution was purified after dialysis with distilled water for 2 days to eliminate the residual reagents, and the crosslinked quaternized chitosan hydrogel solution (CHACC) was obtained. The acetyl salicylate solution in varying proportions (the molar ratio of CHACC to acetyl salicylate in CHACA-1 was 1:0.75, CHACA-2 was 1:1, CHACAA-3 was 1:1.25, CHACAA-4 was 1:1.5, CHACAA-5 was 1:1.75, CHACA-6 was 1:2) was added to the solution and stirred magnetically for 1 h, until well mixed. Afterwards, 0.2 mL of glycerol was added and stirred continued for 12 h. The solution was then ultrasonically defoamed with an ultrasonic cleaner for 30 min. Finally, the hydrogel solution was poured into a Teflon sheet mold and dried at 40 °C for 48 h to obtain the CHACAA hydrogel films. The detailed preparation method of CHACAA was shown in Figure 1.

### 3.3. Characterization of Hydrogel Films

#### 3.3.1. ^1^H Nuclear Magnetic Resonance Spectroscopy

The ^1^H NMR analysis was carried out using a Bruker AVIII-500 Spectrometer (500 MHz, Fällanden, Switzerland) at 25 °C. Derivatives CS, HACC, and HACAA were dissolved in D_2_O, while CHACAA hydrogel films were dissolved in DMSO-d6 at a concentration of 10 mg/mL for further NMR measurement.

#### 3.3.2. Scanning Electron Morphology

The transverse section of hydrogel films fractured in liquid nitrogen was sputtered with gold and detected by SEM S-4800 instrument (Hitachi, Tokyo, Japan). Measurements were carried out using an In-Lens detector at 5 kV.

#### 3.3.3. Thermal Analysis

The thermal stability of hydrogel films was analyzed by a thermogravimetric instrument (Mettler 5 MP, Mettler-Toledo, Greifensee, Switzerland) at 40–500 °C. All specimens were heated at 10 °C/min and under a nitrogen flow of 20 mL/min to obtain derivative thermogravimetric (DTG) data and thermogravimetric analysis (TGA).

#### 3.3.4. Thickness and Density

The thickness of hydrogel films was measured using a digital micrometer (Jingcheng, China) with a 0.001 mm precision. The values obtained from each sample at five different locations were averaged and then used for subsequent volume calculations. The density of each hydrogel film was calculated from the ratio of weight to volume.

#### 3.3.5. Mechanical Properties

The tensile strength (TS) and elongation at break (EB) were determined on a universal tensile testing machine (Instron 5848 MicroTester, High Wycombe, UK). The hydrogel films were mounted between the grips of the machine and operated with a 100 N load cell at a cross-head speed of 1 cm/min. The values of EB (%) and TS (MPa) were determined from the resulting stress–strain curves. All measurements were performed in triplicate.

#### 3.3.6. Water Content, Swelling Degree, and Water Solubility

Firstly, the hydrogel films were weighed for initial weight (*W*_0_). Subsequently, the film samples were dried at 75 °C for 24 h in a hot oven, and the dry weight (*W*_1_) was recorded. Then, the dried films were immersed in 30 mL of phosphate buffered saline (PBS) solution (pH 7.4) at room temperature. After 24 h, the swelled films were taken out and weighed after gently drying with filter paper (*W*_2_). Finally, the films were dried again at 75 °C until constant weight, to define dry mass (*W*_3_). The measurement of each film was performed in triplicate. Water content, swelling degree, and water solubility were calculated as follows:(1)$$Watercontent\left(\%\right)=(W_{0}-W_{1})/W_{0}\times 100$$
(2)$$Swellingdegree\left(\%\right)=(W_{2}-W_{1})/W_{1}\times 100$$
(3)$$Watersolubility\left(\%\right)=(W_{1}-W_{3})/W_{1}\times 100$$

#### 3.3.7. In Vitro Degradation

The dried hydrogel films were immersed in 10 mL of PBS solution (containing trypsin with a concentration of 0.1 mg/mL) and then shaken in a constant temperature oscillating incubator (100 rpm, 37 °C). After 48 h, hydrogel films were taken from the medium and washed with distilled water. Later, the hydrogel films were dried until a constant weight of each fraction was achieved. The in vitro degradation was measured in terms of weight remaining ratio, which was calculated by the following equation:(4)$$Weight\;remaining\;ratio\left(\%\right)=(1-W_{f}/W_{i})\times 100$$
where *W_i_* is the initial weight of the hydrogel film and *W_f_* is the final weight after degradation.

### 3.4. In Vitro Release Study

The in vitro release behavior of acetylsalicylic acid is a vital index to evaluate the practicability of AA-loaded hydrogel films. In this study, the dialysis method was used to investigate the release performance of CHACAA hydrogel films with the aid of a dissolution tester (RC1207DP, Tianjin, China) and UV-vis spectrophotometer (T6, Pgeneral, Beijing, China). During the experiment, 500 mL of phosphate buffered saline (PBS, pH = 7.4) buffer solution was used as the dissolving medium to simulate the environment in vivo. Briefly, a quarter mass of each hydrogel film, as well as 20 mL of PBS medium, were transferred into a dialysis tube (molecular weight cut-off: 8000–12,000 Da) and dialyzed against 480 mL of PBS at 37 °C with the moderate rotation speed of 100 rpm. At predetermined time points (2, 4, 6, 8, 10, 12, 24, and 48 h), 20 mL of release solution was extracted and analyzed spectrophotometrically (300 nm) to calculate the amount of drug released, based on the standard curve of acetylsalicylic acid. After each extraction, the solution was replaced with fresh phosphate buffer solution immediately. The drug release study for AA was also carried out in a similar way. The release percentage was calculated by the ratio of the cumulative release weight to the initial weight.

### 3.5. Evaluation of Antibacterial Activity

The antibacterial activity of the hydrogel films was evaluated by the spread plate counting method [46,47]. Firstly, the sterile hydrogel films (4 × 4 cm) were taken in a petri dish, and then 0.2 mL of bacterial suspension uniformly dispersed in warm agar medium (10^6^ CFU/mL) was evenly applied to the surface of the hydrogel film. The inoculated hydrogel films were incubated at 37 °C for 24 h. Next, the test hydrogel films containing the bacterial medium were placed in 10 mL of phosphate buffer for ultrasonic dispersion for 5 min, and then continuously stirred until the medium was fully dispersed in the buffer. Finally, 0.1 mL of the buffer was inoculated on an agar plate and evenly coated with a sterile glass spreading rod, and then the agar plate was placed in a 37 °C incubator for 24 h. At the same time, a blank control without hydrogel sample was set. After the culture was completed, the growth of bacterial colonies on the solid medium was observed. Three agar plates were coated for each sample as parallel experiments.

### 3.6. Cytotoxicity Assay

#### 3.6.1. Cell Culture and Sample Preparation

Cell culture: RAW 264.7 mouse macrophages were cultured in Dulbecco’s modified Eagle’s medium (DMEM) with 1% penicillin/streptomycin and 10% fetal bovine serum at 37 °C in a CO_2_ environment (5%). When they reached 80% confluence, cells were detached from the bottles using trypsin and plated in 96-well plates for subsequent experiments.

Sample preparation: 2.5 mg of CHACAA hydrogel films were weighed and soaked in 5 mL of complete cell medium for 24 h, and the extracts were taken for cell experiments [48]. All sample solutions were filtered through a sterile needle filter before testing.

#### 3.6.2. Cell Viability Assay

The cytotoxicity of CHACAA hydrogel films on RAW 264.7 mouse macrophage cells was evaluated in accordance with a previous method [37]. Briefly, RAW 264.7 cells inoculated in 96-well plates were cultured in a CO_2_ incubator before treatment. When the density of cells reached 50%, the DMEM medium was replaced by a fresh medium containing samples and incubated for 24 h. Afterwards, the medium of 96-well plates was extracted, and 100 μL of MTT working solution was injected into each well and incubated for another 4 h. Finally, 150 μL of DMSO was added to dissolve the crystals, and the absorbance of each well at 490 nm was measured. Every test step was redone three times and the percentage of viability was calculated as follows:(5)$$Cellviability\left(\%\right)=A_{sample}/A_{control}\times 100$$
where *A_sample_* represents the absorbance of the test sample and *A_control_* refers to the absorbance of the control sample.

### 3.7. Anti-Inflammatory Activity

Anti-inflammatory activity of the hydrogel films, including the inhibitory capacity against nitric oxide and TNF-α inflammatory factor, was carried out in an accordance with the procedure that had been reported [49]. And the specific methods were as follows.

#### 3.7.1. Nitric Oxide Release Assay

The effect of the hydrogel films on nitric oxide (NO) production of RAW 264.7 cells after LPS stimulation was evaluated using the Total Nitric Oxide Assay Kit (Beyotime, Shanghai, China). Briefly, RAW 264.7 cells with a density of 1 × 10^5^ cells/well were seeded onto a 96-well flat-bottom culture plate and cultured for 12 h. Next, the medium was removed, and a fresh medium containing lipopolysaccharide (LPS) and a sample solution were added and cultured for another 24 h. Finally, 80 μL of cell supernatant was collected and reacted with 80 μL of Griess reagent at room temperature for 10 min. The absorbance of each well at 540 nm was detected and the NO level was calculated using NaNO_2_ as a standard (y = 0.00909x + 0.05185, R^2^ = 0.99997).

#### 3.7.2. TNF-α Release Assay

The effect of the hydrogel films on the production of TNF-α in RAW 264.7 cells was also investigated using ELISA kits (Solarbio, Bejing, China). The collection process of RAW 264.7 cell supernatant was followed. Section 3.6.1 and the TNF-α content were determined based on the standard protocols provided in the kits.

### 3.8. Statistical Analysis

All experiments were repeated three times and the descriptive data were presented as the means ± standard deviations. The experimental data on differences between groups were evaluated by One-way ANOVA in SPSS software (version 24.0), and the difference was considered statistically significant if the *p* value was less than 0.05.

## 4. Conclusions

The main purpose of this study was to improve the bioactivity of chitosan-based polymer materials and achieve the sustainable release of acetylsalicylic acid. For this purpose, CHACAA hydrogel films were prepared with HACC as a raw material to realize the application research of chitosan-based hydrogel in medical biomaterials. The main conclusions are as follows: (i) the hydrogel films with correct structural characteristics preserved abundant micropores and ensured the excellent acetylsalicylic acid load capacity through this. (ii) Through the analysis of the physical properties it was found that when the ratio of HACC to AA was 1:1.25 and 1:1.5, the hydrogel films (CHACAA-3 and CHACAA-4) possessed favorable thermal stability and mechanical properties. Notably, the ionic bonding capacity of acetylsalicylic acid in these hydrogel films was used to achieve drug slow-release effects. (iii) The hydrogel films revealed admirable antibacterial activity and anti-inflammatory properties, exhibiting 100% efficacy in inhibiting bacterial growth and a robust ability to reduce inflammatory factors by over 60%. In summary, this study provides evidence that chitosan quaternary ammonium salt containing acetylsalicylic acid is a highly effective antibacterial and anti-inflammatory agent, especially when prepared in hydrogel materials, giving it excellent drug delivery ability. The prepared hydrogel could be a suitable candidate for tissue engineering, bio-adhesives, and wound dressing patches applications. Of course, further research is necessary to enhance the efficiency of drug delivery and to expand the range of diseases susceptible to treatment with this material.

## Data Availability

The raw data supporting the conclusions of this article will be made available by the authors on request.

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
