# Peer review of "Preparation and Properties of Crosslinked Quaternized Chitosan-Based Hydrogel Films Ionically Bonded with Acetylsalicylic Acid for Biomedical Materials"

_marinedrugs, 2024, doi:10.3390/md22100450_

Round 1

Reviewer 1 Report

Comments and Suggestions for Authors

Particular comments

1)                  Introduction section, lines 48-49: What do authors mean by “finite biological activity”?

2)                  Results and Discussion, lines 107-108: What do authors mean by “The hydrogel films were developed using the environmentally friendly biomass quaternized chitosan (HACC) as matrix”.

3)                  Results and Discussion, line 109: What do authors mean by “reinforcing filler”?

4)                  Results and Discussion, lines 115-117: How did the authors determine that a portion of the aspirin binds through ionic bonding and another by physical blending?

5)                  For clarity the authors should indicate the hydrogens in the chemical structures of Figure 1.

6)                  Acetylated and deacetylated groups are found in the structure of CS, only the deacetylated unit is shown in Figure 1. Do the authors have a reason for presenting only the deacetylated unit? in which case, it should be made clear that only part of the CS structure is presented.

7)                  The authors should deepen the structural characterization of the films formed using other techniques such as infrared spectroscopy.

8)                  Results and Discussion, line 144, What is the size of the pores formed?

9)                  The authors should provide a more detailed discussion regarding the structural differences between the films obtained.

10)               Results and Discussion, lines 165-166, It is not clear what the authors mean by “It could also be found from Figure 3a that different CHACAA hydro-gel films exhibited different thermogravimetric behavior at this stage”.

11)               Results and Discussion, lines 176-177, because it is obvious that density and thickness should correlate positively with the amount of aspirin.

12)               Results and Discussion, lines 177-178, Why are the mechanical properties key factors affecting the application of hydrogels in biomedical field.

13)               Table 1 shows that the CHAACA-3 film shows values of Tensile strength and Elongation at the break that are outside the general trend of the series and why the authors consider these results satisfactory?

14)               Results and Discussion, lines 200-201, one of the explanations given by the authors for the higher degradation of the films CHACAA-5-6 is probably the larger pore size of the film, however this is not in the manuscript.

15)               Results and Discussion, line 209, to which the authors refer to “uncoated acetylsalicylic”.

16)               Are the proposed films proposed as topical formulations? to which the authors refer to “The in vitro release of uncoated acetylsalicylic acid exhibited a “burst effect”, which might become the source of shortened action time, multiple and repeated dosing, and increased cytotoxicity of liver and kidney”? What is released from the film referred as “uncoated acetylsalicylic acid” and to which the burst effect is attributed.

17)               Results and Discussion, line 217, how do the authors know that aspirin release is mediated by a diffusion mechanism?

18)               Results and Discussion, line218, the hydrogen bonds in the exposed films how they are formed?

19)               The authors should further investigate the different mechanisms involved in aspirin release and relate them to the different structural features of the films.

20)               Results and Discussion, line 239, what the authors refer to as blank in microbiological studies? it should be indicated what type of bacterial strains were used in the study.

21)               Figure 6 should incorporate the statistically significant differences between the groups.

22)               Materials, the CS molecular weight should be indicated.

23)               Conclusions, line 441, in the conclusion the authors refer to nanoscale pores, while in the results and discussion, they refer to micropores, of what size are the pores formed in the structure of the films?

24)               Conclusions, line 445, the authors state in their conclusions that they avoided the burst effect, however, this phenomenon was observed for all films in the release studies.

Author Response

Dear reviewer,

Thank you for your comments concerning our manuscript entitled “Preparation and properties of crosslinked quaternized chitosan-based hydrogel films ionically bonded with acetylsalicylic acid for biomedical materials”. Those comments are all valuable and very helpful for revising and improving our paper. We have studied comments carefully and have made corrections which we hope meet with approval. The main corrections in the manuscript are as following:

  1. Introduction section, lines 48-49: What do authors mean by “finite biological activity”?

Answer: Thank you for your comments. What we mean here is that the biological activity is relatively weak and we have modified it to “weak bioactivity”.

  1. Results and Discussion, lines 107-108: What do authors mean by “The hydrogel films were developed using the environmentally friendly biomass quaternized chitosan (HACC) as matrix”.

Answer: Thank you for your comments. What we mean here is that the main raw material that constitutes the hydrogel films is quaternized chitosan (HACC).

  1. Results and Discussion, line 109: What do authors mean by “reinforcing filler”?

Answer: Thank you for your comments. What we mean here is that acetylsalicylic acid is the primary additive incorporated into the hydrogel films to improve its biological activity.

  1. Results and Discussion, lines 115-117: How did the authors determine that a portion of the aspirin binds through ionic bonding and another by physical blending?

Answer: Thank you for your comments. For hydrogel films CHACAA 3-6, since the molar content of acetylsalicylic acid is higher than that of quaternized chitosan, some acetylsalicylic acid must exist in physical blending except for ionic bonding. As for as hydrogel films CHACAA 1-2, we calculated the degree of substitution (DS) of acetylsalicylic acid incorporated into quaternized chitosan via ionic bonding through deionized water dialysis and elemental analysis. It was calculated that the degrees of substitution of derivatives CHACAA 1-2 were 65.74% and 68.98%, which means that a part of the weakly interacting acetylsalicylic acid was dialyzed away. Therefore, we concluded that one part of acetylsalicylic acid anions was bound to CHACC by ionic bonding, while the other part was physically blended. However, because the DS data does not appear in paper, we deleted this point of view in revised manuscript to avoid misunderstanding.

  1. For clarity the authors should indicate the hydrogens in the chemical structures of Figure 1.

Answer: Thank you for your comments. In general, the representation of methyl and methylene groups in chemical structural formulas can be simplified to improve the readability without losing the accuracy of chemical information. Therefore, the chemical structures of Figure 1 can clearly reflect the position and number of different hydrogen atoms.

  1. Acetylated and deacetylated groups are found in the structure of CS, only the deacetylated unit is shown in Figure 1. Do the authors have a reason for presenting only the deacetylated unit? in which case, it should be made clear that only part of the CS structure is presented.

Answer: Thank you for your kind suggestions and according to your recommendation we have revised the structure of chitosan and chitosan derivatives in Scheme 1 and Figure 1. Specifically, in addition to the chemical structure of chitosan bearing acetylsalicylic acid, the D-glucosamine and N-acetyl-D-glucosamine units can also be observed in Scheme 1 and Figure 1.

  1. The authors should deepen the structural characterization of the films formed using other techniques such as infrared spectroscopy.

Answer: Thank you for your comments. We have performed the fourier transform infrared spectroscopy analysis of the hydrogel films. However, the characteristic peaks of secondary alcohols and aliphatic ethers (1380 and 1040 cm-1) produced by the reaction of HACC with epichlorohydrin were highly overlapping with the characteristic peaks of chitosan itself. It was not possible to determine whether the crosslinking reaction was successful. Therefore, we showed the NMR data that could support the correct structure of all samples in the manuscript.

  1. Results and Discussion, line 144, What is the size of the pores formed?

Answer: Thank you for your kind suggestions and according to your recommendation we have added the description of the size of the pores formed in revised manuscript. That is “In contrast, the hydrogel films exhibited 3D microporous structure after absorbing water, with the macropores being flat in shape and ranging in size from 100 to 400 μm”.

  1. The authors should provide a more detailed discussion regarding the structural differences between the films obtained.

Answer: Thank you for your comments. In fact, the chemical structures of CHACAA hydrogel films remain the same, with the main difference being the varying content of acetylsalicylic acid. Therefore, the main differences between them are primarily reflected in physicochemical properties. And we have provided a detailed discussion of these physicochemical property differences in the revised manuscript.

  1. Results and Discussion, lines 165-166, It is not clear what the authors mean by “It could also be found from Figure 3a that different CHACAA hydro-gel films exhibited different thermogravimetric behavior at this stage”.

Answer: Thank you for your kind suggestions and according to your recommendation we have revised this sentence to “It could also be found that different CHACAA hydrogel films exhibited different thermogravimetric behavior at this stage, that is, the maximum weight loss rates of the hydrogel films gradually decreased with the increased of acetylsalicylic acid content”.

  1. Results and Discussion, lines 176-177, because it is obvious that density and thickness should correlate positively with the amount of aspirin.

Answer: Thank you for your comments. According to the data, the thickness and density were positively correlated with the amount of acetylsalicylic acid. And we gave examples to prove it later.

  1. Results and Discussion, lines 177-178, Why are the mechanical properties key factors affecting the application of hydrogels in biomedical field.

Answer: Thank you for your comments. Mechanical properties are key factors affecting the application of hydrogels in biomedical field since biomedical materials require resistance to stress and stretch-out caused by body movement. We have added this explanation in the revised manuscript.

  1. Table 1 shows that the CHAACA-3 film shows values of Tensile strength and Elongation at the break that are outside the general trend of the series and why the authors consider these results satisfactory?

Answer: Thank you for your comments. Study has shown that the TS and EB of human skin tissue are in the range of 1-32 MPa and 17-207%. Within this range, the higher the values of TS and EB, the stronger the hydrogel films resistance to stress and stretch-out caused by body movement. Therefore, we concluded that CHACAA-3 possessed the best mechanical property.

  1. Results and Discussion, lines 200-201, one of the explanations given by the authors for the higher degradation of the films CHACAA-5-6 is probably the larger pore size of the film, however this is not in the manuscript.

Answer: Thank you for your comments. The pore size of CHACAA 5-6 hydrogel films is really not described in the manuscript. So, we have revised this sentence.

  1. Results and Discussion, line 209, to which the authors refer to “uncoated acetylsalicylic”.

Answer: Thank you for your comments. The “uncoated acetylsalicylic acid” means the pure acetylsalicylic acid. In this study, the release curve of pure acetylsalicylic acid was used as a control, and the drug release assay was performed in the same way as CHACAA hydrogel films containing acetylsalicylic acid. And we have described it in Materials and Methods.

  1. Are the proposed films proposed as topical formulations? to which the authors refer to “The in vitro release of uncoated acetylsalicylic acid exhibited a “burst effect”, which might become the source of shortened action time, multiple and repeated dosing, and increased cytotoxicity of liver and kidney”? What is released from the film referred as “uncoated acetylsalicylic acid” and to which the burst effect is attributed.

Answer: Thank you for your comments. “Uncoated acetylsalicylic acid” does not refer to acetylsalicylic acid released from the hydrogel films. but to pure acetylsalicylic acid that is not loaded into the controlled release material.

  1. Results and Discussion, line 217, how do the authors know that aspirin release is mediated by a diffusion mechanism?

Answer: Thank you for your comments. We came to this conclusion by reviewing the literature (Drug delivery based on a supramolecular chemistry approach by using chitosan hydrogels. Int. J. Bio. Macromol. 2023, 248, 125800; Chitosan nanoparticles loading oxaliplatin as a mucoadhesive topical treatment of oral tumors: Iontophoresis further enhances drug delivery ex vivo. Int. J. Bio. Macromol. 2020, 154, 1265-1275).

  1. Results and Discussion, line218, the hydrogen bonds in the exposed films how they are formed?

Answer: Thank you for your comments. The hydrogen bonds were formed between the exposed hydroxyl group and amino group on CHACAA hydrogel with water molecules.

  1. The authors should further investigate the different mechanisms involved in aspirin release and relate them to the different structural features of the films.

Answer: Thank you for your comments. At present, we have carried out the drug release experiment of CHACAA hydrogel films according to the existing experimental conditions, and analyzed the release behavior through the obtained data and the previous literature. In the future work, we will further study different mechanisms of aspirin release according to your suggestions.

  1. Results and Discussion, line 239, what the authors refer to as blank in microbiological studies? it should be indicated what type of bacterial strains were used in the study.

Answer: Thank you for your comments. The “blank” refers to the “blank control group”. In the manuscript, we revised it and added the numbers of bacterial strains.

  1. Figure 6 should incorporate the statistically significant differences between the groups.

Answer: Thank you for your kind suggestions and according to your recommendation we have added the statistically significant difference analysis in Figure 6.

  1. Materials, the CS molecular weight should be indicated.

Answer: Thank you for your kind suggestions and according to your recommendation we have added the information of CS molecular weight.

  1. Conclusions, line 441, in the conclusion the authors refer to nanoscale pores, while in the results and discussion, they refer to micropores, of what size are the pores formed in the structure of the films?

Answer: Thank you for your comments. The pores formed by the hydrogel films range from 100 to 400 μm and are not nanoscale pores. So, we revised it to micropores in the manuscript.

  1. Conclusions, line 445, the authors state in their conclusions that they avoided the burst effect, however, this phenomenon was observed for all films in the release studies.

Answer: Thank you for your comments. In the manuscript, we changed this part of the description to “Notably, ionic bonding capacity of acetylsalicylic acid in these hydrogel films was used to achieve drug slow-release effects”.

The revised manuscript has been submitted to journal. We hope that the responses and the revised manuscript adequately address your concerns. Thank you for your time and concerns.

Reviewer 2 Report

Comments and Suggestions for Authors

Below are my comments on the article entitled “Preparation and properties of crosslinked quaternized chitosan-based hydrogel films ionically bonded with acetylsalicylic acid for biomedical materials” submitted to Mar. Drugs.

1.      In the abstract, revise the open name of (CHACAA). I propose to add the word “crosslinked” after the word “six”. Also, the open name of the abbreviations (HACC) should be written at this first mention.

2.      The introduction should be strengthened by displaying previous works on utilizing cationic chitosan as drug delivery carriers and discussing the gap which is covered by your work.

3.      Line 113, “different doses of acetylsalicylate” correct to acetylsalicylic acid as mentioned in the scheme.

4.      I see that the TGA curves did not show significant change in the thermal stabilities of CHACAA3-6. It would be better to re-discuss these curves with respect to the initial decomposition temperature, as well as the weight loss (%) at different temperatures intervals.   

5.  CHACAA-3 gave the highest TS and EB, it would be good if you explain the reason behind this result.

6.    The In vitro release and anti-inflammatory results need to be compared with those reported for other acetylsalicylic loaded hydrogel films in the literature.

7.      Provide the molecular weight of chitosan used.

8.      Provide the magnification of the SEM, in the text or in the Fig. caption.

9.      The references of the Cell viability assay, Nitric oxide release assay, and TNF-α release assay are missing.

Author Response

Dear reviewer,

Thank you for your comments concerning our manuscript entitled “Preparation and properties of crosslinked quaternized chitosan-based hydrogel films ionically bonded with acetylsalicylic acid for biomedical materials”. Those comments are all valuable and very helpful for revising and improving our paper. We have studied comments carefully and have made corrections which we hope meet with approval. The main corrections in the manuscript are as following:

  1. In the abstract, revise the open name of (CHACAA). I propose to add the word “crosslinked” after the word “six”. Also, the open name of the abbreviations (HACC) should be written at this first mention.

Answer: Thank you for your kind suggestions and according to your recommendation, we have added the word “crosslinked” after the word “six” and mentioned the abbreviations (HACC) first in Abstract.

  1. The introduction should be strengthened by displaying previous works on utilizing cationic chitosan as drug delivery carriers and discussing the gap which is covered by your work.

Answer: Thank you for your comments and according to your recommendation, we have revised the Introduction part.

  1. Line 113, “different doses of acetylsalicylate” correct to acetylsalicylic acid as mentioned in the scheme.

Answer: Thank you for your comments and according to your recommendation, we have revised acetylsalicylate to acetylsalicylic acid.

  1. I see that the TGA curves did not show significant change in the thermal stabilities of CHACAA3-6. It would be better to re-discuss these curves with respect to the initial decomposition temperature, as well as the weight loss (%) at different temperatures intervals.

Answer: Thank you for your comments. Although CHACAA 3-6 hydrogel films do not change significantly in TGA, they do change in range of 160°C to 260°C, which can be observed by the DTG curve. And we have given special discussion to this phenomenon in revised manuscript.

  1. CHACAA-3 gave the highest TS and EB, it would be good if you explain the reason behind this result.

Answer: Thank you for your comments and according to your recommendation, we have explained the reason for CHACAA-3 having the highest TS and EB in revised manuscript. This might because the blending of quaternized chitosan and acetylsalicylic acid in this ratio caused polymer chains to have more chain entanglement. And the higher applied tensile force was used to overcome the high amount of polymer chain entanglement, resulting in higher tensile strength and elongation at break.

  1. The In vitro release and anti-inflammatory results need to be compared with those reported for other acetylsalicylic loaded hydrogel films in the literature.

Answer: Thank you for your comments. According to your recommendation, we have analyzed and compared the phenomena observed in our research with other acetylsalicylic loaded hydrogels in the literature in release studies. In anti-inflammatory activity, because of the different expression of the data results, it is not possible to make an intuitive comparison with other hydrogels. Moreover, we set up positive control and blank control in the article, which has a good reference for evaluating the anti-inflammatory activity of CHACAA hydrogel films.

  1. Provide the molecular weight of chitosan used.

Answer: Thank you for your kind suggestions and according to your recommendation we have added the information of chitosan molecular weight.

  1. Provide the magnification of the SEM, in the text or in the Fig. caption.

Answer: Thank you for your kind suggestions and according to your recommendation we have provided the magnification of the SEM in the figure caption.

  1. The references of the Cell viability assay, Nitric oxide release assay, and TNF-α release assay are missing.

Answer: Thank you for your kind suggestions and according to your recommendation we have added the references of the Cell viability assay, Nitric oxide release assay, and TNF-α release assay in revised manuscript.

The revised manuscript has been submitted to journal. We hope that the responses and the revised manuscript adequately address your concerns. Thank you for your time and concerns.

Reviewer 3 Report

Comments and Suggestions for Authors

marinedrugs-3202905. "Preparation and properties of crosslinked quaternized chitosan-based hydrogel films ionically bonded with acetylsalicylic acid for biomedical materials" by Zhanyong Guo et al.  The study focuses on ionic bonding strategy between highly active quaternized chitosan and acetylsalicylic acid. Overall, this manuscript was well-written and constructed. In my opinion, the manuscript is partially suitable for publication in marinesdrugs but does not meet the standard. Meanwhile, before the article can be published, the authors should be carried out the corrections as listed below:
Comment
1. The introduction is written very well and the aim of the study is very clear

2-The characterization using 1HNMR looks nice and discussed very well but the nanoparticles still need more characterization, for example zeta potential (to know about the surface charge), and IR spectra.

3-Authors need to prove whether the acetylsalicylate encapsulated or just on the surface.

4-As a whole, the discussion part is also well written but needs correlation of this work to a previous work.

5- Drug release study part: In which pH media the drug release has been investigated? In order to target the cancer cell, the release media should be acidic and in many cases, also should be acidic with glutathione concentration. For these reasons, authors should do the release in neutral (blood stream) and acidic or acidic with GSH to mimic the cancer cell media.

6: The resolution of the figures needs improvement.

7-The conclusion is also well written and concludes the finding in a good way

Comments on the Quality of English Language

minor editing

Author Response

Dear reviewer,

Thank you for your comments concerning our manuscript entitled “Preparation and properties of crosslinked quaternized chitosan-based hydrogel films ionically bonded with acetylsalicylic acid for biomedical materials”. Those comments are all valuable and very helpful for revising and improving our paper. We have studied comments carefully and have made corrections which we hope meet with approval. The main corrections in the manuscript are as following:

  1. The introduction is written very well and the aim of the study is very clear.

Answer: Thank you for your kind comments.

  1. The characterization using 1H NMR looks nice and discussed very well but the nanoparticles still need more characterization, for example zeta potential (to know about the surface charge), and IR spectra.

Answer: Thank you for your comments. Since we prepared hydrogel films instead of nanoparticles, we did not analyze their zeta potential. We have performed the fourier transform infrared spectroscopy analysis of the hydrogel films. However, the characteristic peaks of secondary alcohols and aliphatic ethers (1380 and 1040 cm-1) produced by the reaction of HACC with epichlorohydrin were highly overlapping with the characteristic peaks of chitosan itself. It was not possible to determine whether the crosslinking reaction was successful. Therefore, we showed the NMR data that could support the correct structure of all samples in the manuscript.

  1. Authors need to prove whether the acetylsalicylate encapsulated or just on the surface.

Answer: Thank you for your kind comments. During the preparation process of hydrogel films, it was inevitable that a small amount of acetylsalicylic acid would adhere on the surface, while the majority of acetylsalicylic acid was encapsulated in the interior. In the hydrogel structure, one part of acetylsalicylic acid anions was bound to CHACC by ionic bonding, while the other part was physically blended. We have proved this viewpoint by calculating the degree of substitution of CHACAA. The calculation results are as shown in the following table.

Compounds

Elemental analyses (%)

Degrees of

Substitution

Deacetylation

(%)

(%)

C

N

H

C/N

CS

41.04

7.39

9.03

5.55

76.25

HACC

CHACAA-1

CHACAA-2

CHACAA-3

CHACAA-4

CHACAA-5

CHACAA-6

39.89

45.72

46.59

43.67

47.36

48.96

48.34

7.42

5.22

5.22

4.86

5.32

5.43

5.36

11.23

10.23

11.26

11.89

10.95

11.09

11.68

5.38

8.76

8.93

8.98

8.90

9.02

9.02

74.63

65.74

68.98

69.97

68.36

70.79

70.65

  1. As a whole, the discussion part is also well written but needs correlation of this work to a previous work.

Answer: Thank you for your kind suggestions and according to your recommendation we have made a contrast with other hydrogel films in Results and Discussion.

  1. Drug release study part: In which pH media the drug release has been investigated? In order to target the cancer cell, the release media should be acidic and in many cases, also should be acidic with glutathione concentration. For these reasons, authors should do the release in neutral (blood stream) and acidic or acidic with GSH to mimic the cancer cell media.

Answer: Thank you for your kind comments. Since the CHACAA hydrogel films are intended to be prepared as medical material for tissue engineering, bio-adhesive, and wound dressing patch applications, the in vitro drug release experiment was performed at pH 7.4 to simulate the normal environment of body fluids. And we have explained the reason for testing drug release behavior of CHACAA hydrogel films only in pH 7.4 environment in section 2.2.

  1. The resolution of the figures needs improvement.

Answer: Thank you for your kind suggestions and according to your recommendation we have improved the resolution of all figures.

  1. The conclusion is also well written and concludes the finding in a good way.

Answer: Thank you for your kind comments.

The revised manuscript has been submitted to journal. We hope that the responses and the revised manuscript adequately address your concerns. Thank you for your time and concerns.

Reviewer 4 Report

Comments and Suggestions for Authors

Authors reported the synthesis and characterization of cross-linked hydroxypropyl trimethyl ammonium chloride chitosan (CHACC) hydrogel with improved mechanical strength and controlled drug release activity. They formulated the non-steroidal anti-inflammatory drug acetylsalicylic acid into CHACC (CHACAA), which exhibited enhanced antibacterial and anti-inflammatory properties. The controlled drug release ability is mediated by electrostatic interactions between CHACC and salicylic acid, the drug used in this study. This newly designed hydrogel showed controlled drug release properties under in vitro conditions and antibacterial/anti-inflammatory activities. The cross-linking steps optimized here are useful for designing chitosan-based hydrogel formulations for drugs in general. All the conclusions are consistent with the arguments presented, and all the references are appropriate to the work.

I have the following comments

1) Increased acetylsalicylic acid does cause approximately 20% cell toxicity, as shown in Fig. 6a. This should be discussed. Subsequently, this change needs to be considered in determining the NO and TNF-α levels.

2) Testing the slow-release formulation in a suitable model organism would be useful to show the application of polymer in vivo.

3) Increasing the font size of the text on all the Figures is required.

4) Authors should include some details in all the Figure options.

5) Delete the spectral region after 9 ppm. Full spectra can be added to the SI file if needed.

6) Which NMR samples are made in D2O/DMSO. Provide this information in methods and also in the Fugure caption.

7) Which version of the hydrogel is used in the presented image in Figure 5? Mention it in the caption.

8) Did the authors use 2D NMR measurements to assign the NMR peaks? Describe it.

9) Is there any NMR data supporting the structural arrangement of acetylsalicylic acid in the CHACAA shown in the Schematic?

Author Response

Dear reviewer,

Thank you for your comments concerning our manuscript entitled “Preparation and properties of crosslinked quaternized chitosan-based hydrogel films ionically bonded with acetylsalicylic acid for biomedical materials”. Those comments are all valuable and very helpful for revising and improving our paper. We have studied comments carefully and have made corrections which we hope meet with approval. The main corrections in the manuscript are as following:

  1. Increased acetylsalicylic acid does cause approximately 20% cell toxicity, as shown in Fig. 6a. This should be discussed. Subsequently, this change needs to be considered in determining the NO and TNF-α levels.

Answer: Thank you for your kind suggestions and according to your recommendation, we have focused discussion on this phenomenon in 2.4 part.

  1. Testing the slow-release formulation in a suitable model organism would be useful to show the application of polymer in vivo.

Answer: Thank you for your comments. At present, we have carried out the drug release experiment of CHACAA hydrogel films according to the existing experimental conditions, and analyzed the release behavior through the obtained data and the previous literature. In the future work, we will further test the slow-release formulation in a suitable model organism that can show the application of polymer in vivo according to your suggestions.

  1. Increasing the font size of the text on all the Figures is required.

Answer: Thank you for your kind suggestions and according to your recommendation, we have increased the font size of all figures.

  1. Authors should include some details in all the Figure options.

Answer: Thank you for your kind suggestions and according to your recommendation, we have added details in all figure options.

  1. Delete the spectral region after 9 ppm. Full spectra can be added to the SI file if needed.

Answer: Thank you for your kind suggestions and according to your recommendation, we have deleted the spectral region after 9 ppm in Fig. 1.

  1. Which NMR samples are made in D2O/DMSO. Provide this information in methods and also in the Figure caption.

Answer: Thank you for your kind suggestions and according to your recommendation, we have provided this information in methods and the Figure caption.

  1. Which version of the hydrogel is used in the presented image in Figure 5? Mention it in the caption.

Answer: Thank you for your kind suggestions and according to your recommendation, we have mentioned the version of the hydrogel used in the presented image shown in Figure 5.

  1. Did the authors use 2D NMR measurements to assign the NMR peaks? Describe it. Is there any NMR data supporting the structural arrangement of acetylsalicylic acid in the CHACAA shown in the Schematic?

Answer: Thank you for your comments. At present, we did not use 2D NMR measurements to assign the NMR peaks. And the obtained NMR data can only prove the correctness of CHACAA structure. In the following work, we will adopt 2D NMR measurements to assign the NMR peaks and analysis the structural arrangement of acetylsalicylic acid in CHACAA.

The revised manuscript has been submitted to journal. We hope that the responses and the revised manuscript adequately address your concerns. Thank you for your time and concerns.

Round 2

Reviewer 1 Report

Comments and Suggestions for Authors

The authors correctly answered all the concerns and made the changes suggested by this reviewer.

Reviewer 3 Report

Comments and Suggestions for Authors

authors have considered most of the comments and the manuscript has significantly improved and now may be publishable in the marine drugs 

Comments on the Quality of English Language

Minor editing is required